# Variability of the Cervical Branch Depending on the Facial Nerve Branching Pattern and Anthropometric Type of the Head

Angela Babuci [1], Victor Palarie [1], Ilia Catereniuc [1], Zinovia Zorina [1], Sergiu Visnevschi [1], Diana Heimes [2,*], Sofia Lehtman [3] and Peer W. Kämmerer [2]

1   Department of Anatomy and Clinical Anatomy, Nicolae Testemitanu State University of Medicine and Pharmacy, 2004 Chișinău, Moldova; angela.babuci@usmf.md (A.B.); victor.palarie@usmf.md (V.P.); ilia.catereniuc@usmf.md (I.C.); zinovia.zorina@usmf.md (Z.Z.); sergiu.visnevschi@usmf.md (S.V.)
2   Department of Oral and Maxillofacial Surgery, University Medical Center Mainz, 55131 Mainz, Germany; peer.kaemmerer@unimedizin-mainz.de
3   Department of Oral and Maxillofacial Surgery and Oral Implantology, Nicolae Testemitanu State University of Medicine and Pharmacy, 2004 Chișinău, Moldova; sofia.lehtman@usmf.md
*   Correspondence: diana.heimes@unimedizin-mainz.de

**Abstract:** (1) Background: Considering that the specialty literature supplies only general data about the variability of the cervical branch of the facial nerve, this study aimed to determine this branch's variation and individual peculiarities depending on the nerve branching pattern and anthropometric type of the head. (2) Methods: The study was conducted on 75 hemifaces of adult formalized cadavers. Ahead of anatomical dissection, each head was measured to establish the anthropometric type, according to Franco and colleagues. The branching patterns were then distributed according to the Davis classification. (3) Results: The number of cervical branches (CB) of the facial nerve varied from one to five branches, with the following rate: 1 CB (61.3%), 2 CB (28%), 3 CB (6.7%), 4 CB (2.7%), and 5 CB (1.3%). Seven branching patterns of the facial nerve were revealed: Type I in 18.7%, Type II in 14.7%, Type III in 20%, Type IV in 14.6%, Type V in 5.3%, Type VI in 18.7%, and Type NI in 8% (bizarre types). According to the branching pattern, the mean numbers of the cervical branches were as follows: Type I—1.6 ± 1.02; Type II—1.4 ± 0.50; Type III—1.4 ± 0.50; Type IV—1.4 ± 0.67; Type V—2.0 ± 1.41; Type VI—1.8 ± 1.12; and Type-NI—1.8 ± 0.75; $p = 0.599$. According to the anthropometric type of the head, the mean number of CB in the mesocephalic type (MCT) was 1.5 ± 0.82, in the dolichocephalic type (DCT), 1.7 ± 0.87, and in the brachycephalic type, (BCT) 1.8 ± 1.04; $p = 0.668$. (4) Conclusions: The cervical branch of the facial nerve varies depending on the facial nerve branching pattern and the anthropometric type of the head. The highest degree of variation was characteristic of BCT and Type V and the lowest, of MCT and Types II, III, and IV.

**Keywords:** cervical branch; facial nerve; variation; connections; anthropometric type

## 1. Introduction

Over the last decades, along with the increasing number of head and neck malignancies accounting for one of the most common cancers worldwide (380,000 cases per year), many surgical interventions to the neck have been performed. Taken together with surgery of the salivary glands, head and neck traumas, and a high demand for rejuvenation procedures and aesthetic surgery, the risk of irreversible facial nerve lesions is high [1–3].

The incidence of facial nerve iatrogenic lesions in routine facial surgery is about 0.1% to 0.5%, while the rate of those lesions in aesthetic surgery varies between 1% and 20% [4]. According to data reported by Li et al. [5], anterograde dissection during regional parotidectomy results in a rate of 4.3%, while the rate of the iatrogenic lesions in retrograde dissection has been reported to be as high as 20.6%. In parotid tumor ablation, the microtraumas of the facial nerve branches may occur due to excessive extension, resulting in postoperative paresis [6]. A systematic review reported a 13% risk of marginal mandibular nerve damage

in both radical-modified and selective neck dissection [7]. Only a few studies have focused on the cervical branch of the facial nerve under the assumption that it innervates only the platysma. The superior fibers of the platysma muscle pass above the mandibular margin, and laterally, they are interconnected with the masseteric fascia. Some fibers of the platysma are medially crossing or overlapping with the opposite fellow and with the depressor anguli oris, risorius, and depressor labii inferioris muscles. Despite the common opinion that the scarification of the cervical branch does not affect patients' life quality, the partial innervation of the lower lip by the cervical branch is reported in the current literature [8,9], and thus, the scarification of the cervical branch of the facial nerve can lead to mimicry impairment. In 1964, De Sousa et al. demonstrated the importance of platysma muscle contraction in lowering the labial commissure and lower lip, which allows a balance in elevating and depressing forces [9]. Lesions to the cervical branch often induce asymmetry due to reduced counterbalancing, causing a disfigurement often mistaken for a marginal mandibular branch lesion [9]. According to Chowdhry et al. [10], the motor innervation of the platysma muscle is clinically significant because in functional disorders of the cervical branch, patients suffer from hyperkinetic motility with aesthetic and motor impairments. The platysma muscle has an important functional role in the protection of the anterior jugular veins from compression and collapsing, and under its contraction, it assures the venous drainage from the anterior jugular veins, located in the interaponeurotic suprasternal space; thus, its scarification or usage in motor nerves reconstruction should be very carefully considered.

Taking into account the risk of permanent nerve damage and the high negative impact of functional and cosmetic impairments on the quality of life among these patients, a preoperative depiction of the branching patterns would be desirable.

The central segments of the facial nerve can be depicted through magnetic resonance imaging (MRI) and computed tomography (CT). Distal to the mastoid segment, direct imaging is not possible due to the thinness of the peripheral nerve. In such cases, indirect landmarks can be used to locate the course and plane of the nerve [11]. The orientation and integration of the white matter tracts and cranial nerve course within the brain can also be examined using tractography in vivo [12–15]. Still, unfortunately, it is not applicable for the extracranial branches of the facial nerve, except for 3D constructive interference in steady-state MRI, which is feasible for the previsualization of the facial nerve trunk and its primary divisions [16]. Recent studies have shown that the intratemporal facial nerve's main trunk and divisions, and even the secondary branches, may be visualized on high-resolution 3 Tesla MRI [17].

Knowledge of the anatomical conditions and common variants of branching patterns becomes increasingly important without sufficient diagnostic imaging modality.

Since 1862, the peripheral course of the facial nerve has been an object in many studies [18,19]. Anatomists have classified different nerve patterns observed during dissection. In 1956, Davis et al. proposed six different patterns according to the presence or absence of the connections between terminal branches based on the dissection of 350 hemifaces (I–VI) [20]. According to Babuci et al. [21], the course of the facial nerve trunk, along with the descending direction, may have a horizontal and even an ascending one, which increases the risk of facial nerve iatrogenic lesions, especially of those extracranial branches that derive from its cervicofacial division. Anatomy books, and even *Terminologia anatomica*, describe the cervical branch of the facial nerve as being solitary [22].

Nevertheless, it is subject to numerical, topographical, and connection variants [23–29]. In the current literature, the cervical branch has been reported to be a single branch (15%), a double branch (55%), or even three branches (30%) [29]. Variants of the platysma muscle innervation by the greater auricular nerve and the cervical branch variability have also been reported [30].

Taking into consideration the limited research on cervical branch variability and the fact that knowledge about its morphological peculiarities is of clinical significance and can be obtained only through anatomical dissection or from head and neck surgical interven-

tions, the goal of this study was to determine the variability and individual morphological peculiarities of the cervical branch depending on the facial nerve branching pattern and anthropometric type of the head.

## 2. Materials and Methods

The study was conducted at the Department of Anatomy and Clinical Anatomy of the Nicolae Testemitanu State University of Medicine and Pharmacy in the Republic of Moldova from 2014 to 2022.

The research was carried out on 75 hemifaces of adult formalized cadavers (59 male and 16 female). Before dissection, the head and neck areas of each corpse were thoroughly examined, and only those hemifaces with intact soft tissues were included in the study.

Ahead of dissection, the measurements of the transverse and longitudinal diameters of the head were taken to establish the anthropometric type of the head. The transverse diameter was measured between two eurions, and the longitudinal one was equal to the distance between the glabella and opistocranion. The cephalic index was calculated according to the following formula:

$$\text{Cepahlic index} = \frac{Transverse\ diameter \times 100}{Longitudinal\ diameter} \tag{1}$$

A total of 78.7% of the analyzed hemifaces were male, and 21.3% were female. The ratio of the right-side specimens was 53.3%, and that of the left ones was 46.7%. In male cadavers, the right hemifaces constituted 49.2% and the left, 50.8%. In females, the right hemifaces were represented by 37.5% and the left ones by 62.5%.

Anthropometrical classification of the heads was conducted according to Franco F. et al. [31] Thus, heads with a cephalic index up to 74.9 were classified as being of dolichocephalic type (DCT), the heads with a cephalic index between 75.0 and 79.9 as mesocephalic type (MCT), and the heads with a cephalic index higher than 80.0 as brachycephalic type (BCT).

As a result of the performed dissections, 14 types of facial nerve branching patterns were established. The branching patterns were distributed according to Davis's classification (Figures 1 and 2) [20]. Davis proposed six different branching patterns based on the presence of connections between the terminal branches of the facial nerve [29]. Those patterns that entirely corresponded to the types identified by Davis et al. [20] were classified as classical branching variants and all the other patterns as atypical. One type was termed Type-NI, which included all the bizarre types that could not be identified in the specialty literature.

For the dissection of the cervical branch, a midline incision was performed on the neck, followed by retrograde dissection to separate the skin and subcutaneous tissue toward the lateral margin of the trapezius muscle. Subsequently, the platysma muscle, situated within the superficial cervical fascia of the neck, underwent retrograde dissection, and its inferior margin was detached from the clavicle. To mitigate the risk of damage to the marginal mandibular branch, facial artery, and vein, the platysma was meticulously dissected along the base of the mandible.

In the majority of dissected hemifaces, the cervical branch was consistently situated medially to the platysma muscle. However, in some instances, the cervical branch penetrated the muscle from the deep plane to the superficial plane in close proximity to the mandibular margin. Subsequently, the nerve followed a descending trajectory on the external surface of the platysma muscle, covering a distance of 3 to 5 cm before connecting to the transverse nerve of the neck.

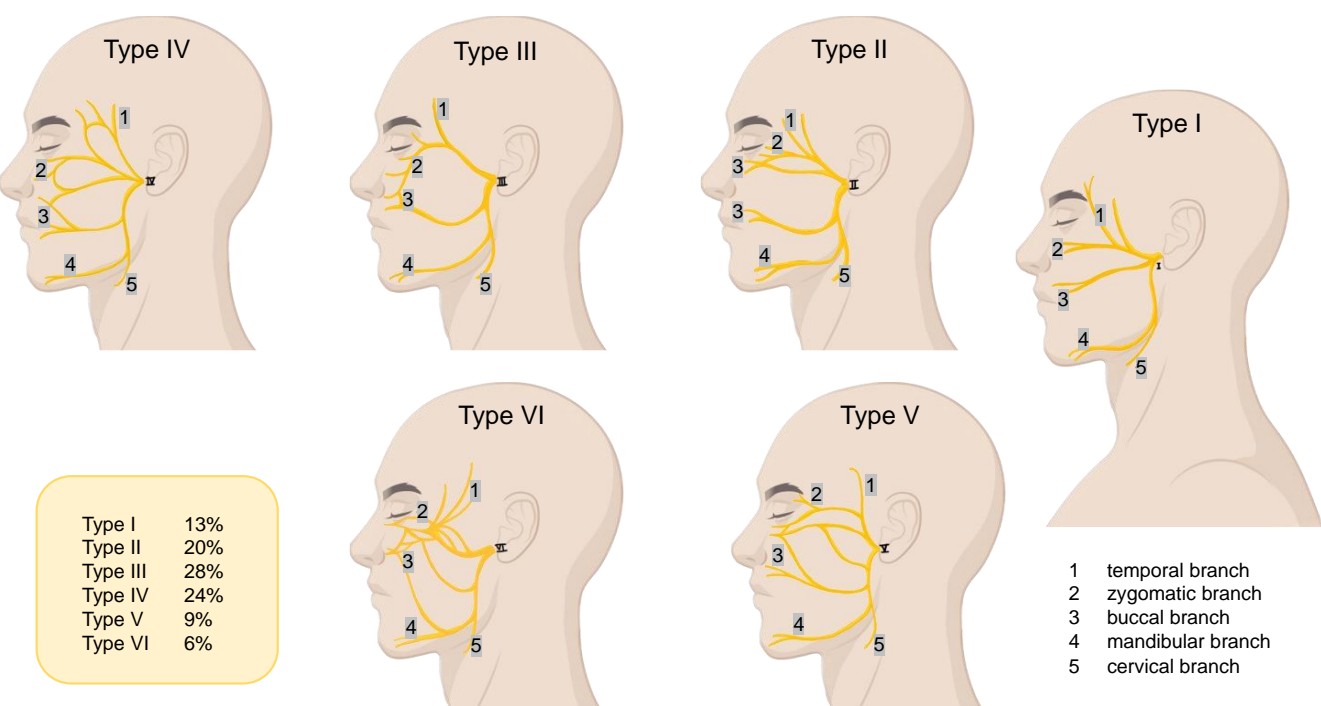

**Figure 1.** The facial nerve branching patterns, according to the Davis classification [20]. Type I: No connection between the temporofacial division and cervicofacial division; Type II: Connections only between the branches of the temporofacial division; Type III: Single connection between the branches of the temporofacial and cervicofacial divisions; Type IV: Combination of type II and III; Type V: Double connection between the branches of the temporofacial and cervicofacial divisions; Type VI: Complex numerous connections between the two divisions wherein the buccal branch receives many fibers from the mandibular branch and cervicofacial division. The values listed indicate the frequency of each type of branching in the population.

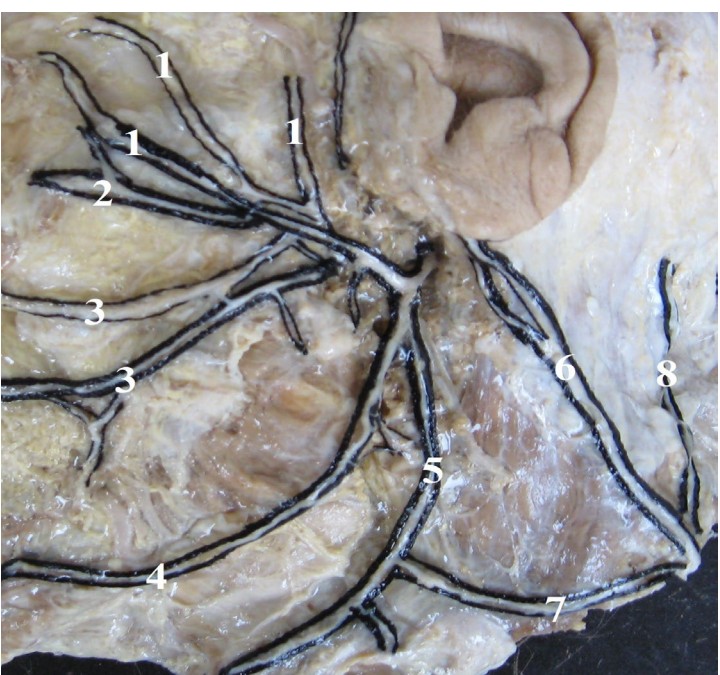

**Figure 2.** Type I of the facial nerve branching pattern: 1—temporal branches; 2—zygomatic branches; 3—buccal branches; 4—marginal mandibular branch; 5—cervical branch; 6—greater auricular nerve; 7—connection of the cervical branch of the facial nerve with the transverse cervical nerve of the cervical plexus; 8—lesser occipital nerve.

Given the elevated risk of cervical branch lesions during retrograde dissection, a deliberate approach was adopted wherein the identification of the facial nerve trunk or its bifurcation angle took precedence. The cervical branch was then dissected along the cervicofacial division of the facial nerve. A combination of soft, cartilaginous, bony, and projection landmarks was employed for surgical access to the facial nerve trunk. These landmarks included the intertragic notch, the triangular cartilaginous prominence of the external acoustic meatus, the insertion point of the anterior margin of the sternocleidomastoid muscle, the digastric muscle, the apex of the mastoid process, the posterior margin of the mandibular ramus, and the mandibular angle. Anterograde dissection demonstrated a higher level of safety in comparison to retrograde dissection.

The Microsoft Excel 2016 processing program functions STDEV and CONFIDENCE, $\chi^2$ test, and one-way ANOVA for comparing the means of three or more independent samples were used to analyze the quantitative and qualitative variables statistically. The same observer took all the measurements and performed the accounting of the cervical branches.

The research project was approved by the Ethics Committee of Nicolae Testemitanu State University of Medicine and Pharmacy in the Republic of Moldova (minute No. 1 of 19 September 2014). The study was conducted in full accordance with the Declaration of Helsinki.

For a relevant statistical analysis, the atypical variants were revisited and, according to general features of the branching pattern, were added to the corresponding classical types, resulting in seven branching types.

The cadavers belonged to the Department of Anatomy and Clinical Anatomy of Nicolae Testemitanu State University of Medicine and Pharmacy in the Republic of Moldova.

## 3. Results

### 3.1. Branching Patterns

Seven branching patterns of the facial nerve were established in the current study: Type I in 18.7%, Type II in 14.7%, Type III in 20%, Type IV in 14.6%, Type V in 5.3%, Type VI in 18.7%, and Type-NI in 8%.

### 3.2. Cephalic Index

A total of 77.3% of hemifaces were classified as MCT, 12% as DCT, and 10.7% as BCT. Specimens with MCT showed a male/female ratio of 81.3%/62.5%, specimens with DCT of 11.9%/12.5%, and BCT of 6.8%/25%. The ratio of the right/left samples in MCT was 77.2%/77.5%; in DCT, 11.4%/12.5%; and in BCT, 11.4%/10%. The distribution of the cephalic index depending on the branching pattern is given in Table 1.

**Table 1.** The mean values of the cephalic index depending on the branching pattern.

| Type of Branching | Mean Value $\pm$ SD | CI 95% |
|:---:|:---:|:---:|
| Type I | 76.6 $\pm$ 1.74 | 75.7–77.5 |
| Type II | 77.1 $\pm$ 2.08 | 75.9–78.4 |
| Type III | 77.3 $\pm$ 1.58 | 76.4–78.1 |
| Type IV | 78.3 $\pm$ 1.52 | 77.4–79.2 |
| Type V | 78.8 $\pm$ 1.76 | 77.0–80.5 |
| Type VI | 77.0 $\pm$ 1.46 | 76.3–77.8 |
| Type-NI | 75.9 $\pm$ 1.35 | 74.8–77.0 |

Note: SD—standard deviation; CI 95%—confidence interval.

The number of cervical branches of the facial nerve varied from one to five branches. The percentages of the cervical branch number variation were as follows: 1 CB (61.3%), 2 CB (28%), 3 CB (6.7%), 4 CB (2.7%), and 5 CB (1.3%) (Figure 3).

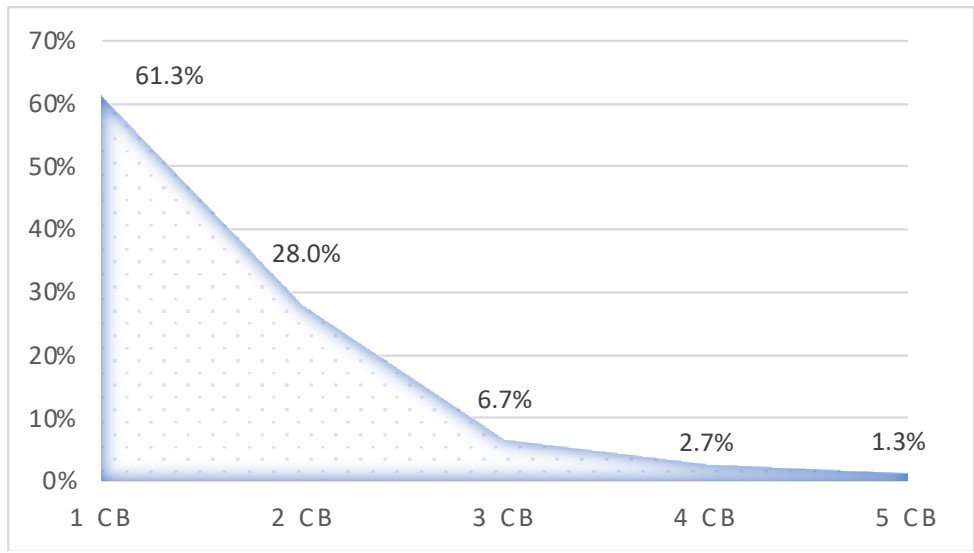

**Figure 3.** Numerical variation in the cervical branch.

In males, the mean value of the cervical branch was 1.6 CB, and in females, 1.5 CB ($p = 0.805$). On the right hemifaces, male individuals showed 1 to 4 CB; on the left, there were 1 to 5 CB. In female cadavers, the number varied from 1 to 4 CB on the right hemifaces and 1 to 2 CB on the left side. The mean value of the cervical branches on the right hemifaces was 1.7 CB, and on the left ones, 1.5 CB ($p = 0.291$). The classical branching pattern had a mean of 1.5 CB, and the atypical branching pattern had a mean value of 1.6 CB ($p = 0.510$).

The $\chi^2$ test demonstrated a statistically significant correlation between the branching type and the presence of the atypical branching pattern ($p = 0.01$; Table 2).

**Table 2.** The ratios of the classical and atypical facial nerve branching patterns.

| Branching Types | Individual Variability | | | | Total |
|---|---|---|---|---|---|
| | Classical Variant of Branching Pattern | | Atypical Variant of Branching Pattern | | |
| | Number of Samples | % | Number of Samples | % | |
| Type I | 8 | 19.5 | 6 | 17.6 | 14 |
| Type II | 3 | 7.3 | 8 | 23.5 | 11 |
| Type III | 8 | 19.5 | 7 | 20.6 | 15 |
| Type IV | 8 | 19.5 | 3 | 8.8 | 11 |
| Type V | 2 | 4.9 | 2 | 5.9 | 4 |
| Type VI | 12 | 29.3 | 2 | 5.9 | 14 |
| Type-NI | 0 | 0.0 | 6 | 17.6 | 6 |
| Total | 41 | 100 | 34 | 100 | 75 |

The variation in the cervical branch number, depending on the facial nerve branching pattern (Table 3), had a frequency of the intragroup variation of 0.767 (df = 6, $p = 0.599$).

**Table 3.** Variation in the cervical branch number depending on the branching pattern.

| Type of Branching | Mean Value ± SD | CI 95% |
|---|---|---|
| Type I | 1.6 ± 1.02 | 1.0–2.1 |
| Type II | 1.4 ± 0.50 | 1.1–1.7 |
| Type III | 1.4 ± 0.50 | 1.1–1.6 |
| Type IV | 1.4 ± 0.67 | 1.0–1.8 |
| Type V | 2.0 ± 1.41 | 0.6–3.4 |
| Type VI | 1.8 ± 1.12 | 1.2–2.4 |
| Type-NI | 1.8 ± 0.75 | 1.2–2.4 |

Note: SD—standard deviation; CI 95%—confidence interval.

In specimens with a mesocephalic type, the mean number of the cervical branches was 1.5 ± 0.82 (CI 95% 1.3–1.7); with a dolichocephalic type, there were 1.7 ± 0.87 (CI 95% 1.1–2.2) branches, and in the brachycephalic type, 1.8 ± 1.04 (CI 95% 1.0–2.5) (Table 4). The frequency of the intragroup variation was 0.406 (df = 2, *p* = 0.668).

**Table 4.** Variation in the cervical branch number depending on the anthropometric type of the head.

| Anthropometric Type of the Head | Mean Value ± SD | CI 95% |
| --- | --- | --- |
| Mesocephalic type | 1.5 ± 0.82 | 1.3–1.7 |
| Brachycephalic type | 1.8 ± 1.04 | 1.0–2.5 |
| Dolichocephalic type | 1.7 ± 0.87 | 1.1–2.2 |

Note: SD—standard deviation; CI 95%—confidence interval.

The current study revealed several variants of the cervical branch such as a solitary cervical branch (Figure 4A), a double cervical branch (Figure 4B–D), a triple cervical branch (Figure 4E–G), and multiple branches (Figure 4H,I). A triple connection between the anterior and posterior cervical branches was found in the case of a double cervical branch (1.3%). The upper connection formed immediately below the point of the cervical branches' origin, and the retromandibular vein was located medially to that connection. The upper part of the external jugular vein was located medially to the middle link and laterally to the inferior one (Figure 4C). In some cases, when there was more than one cervical branch, they were connected by double (Figure 4I), triple (Figure 4C), and multiple (Figure 4H) parallel connections. In other cases, wide, narrow, loop-shaped connections between the cervical branches were observed (Figure 4D,F).

On the external surface of the face, the highest level of the cervical branch course was along the mandibular margin. In three cases (5.3%), one of the cervical branches was located beneath the mandible and higher than its margin (Figure 4C). The cervical branch was connected to the marginal mandibular branch in 24% of cases, among which a single connection was determined in 20% of cases, double connections in 1.3%, and multiple connections in 2.7%. Connections between the cervical branch and the greater auricular nerve were observed in 38.7% of cases, of which double connections constituted 8%, and in 1.3% of cases, triple connections were observed. In all hemifaces, connections between the cervical branch and the transverse nerve of the neck were shown.

In two cases (2.7%), the cervical branch formed a loop around the retromandibular vein and followed its path posteriorly to the retromandibular vein, forming connections with the neck's greater auricular and transverse nerves.

Five types of connections were highlighted between the cervical branch and the greater auricular nerve:

1. Type I—a single connection (42.7%);
2. Type II—two close connections (8%);
3. Type III—two distant connections (12%);
4. Type IV—three connections (5.3%);
5. Type V—no connections (32%).

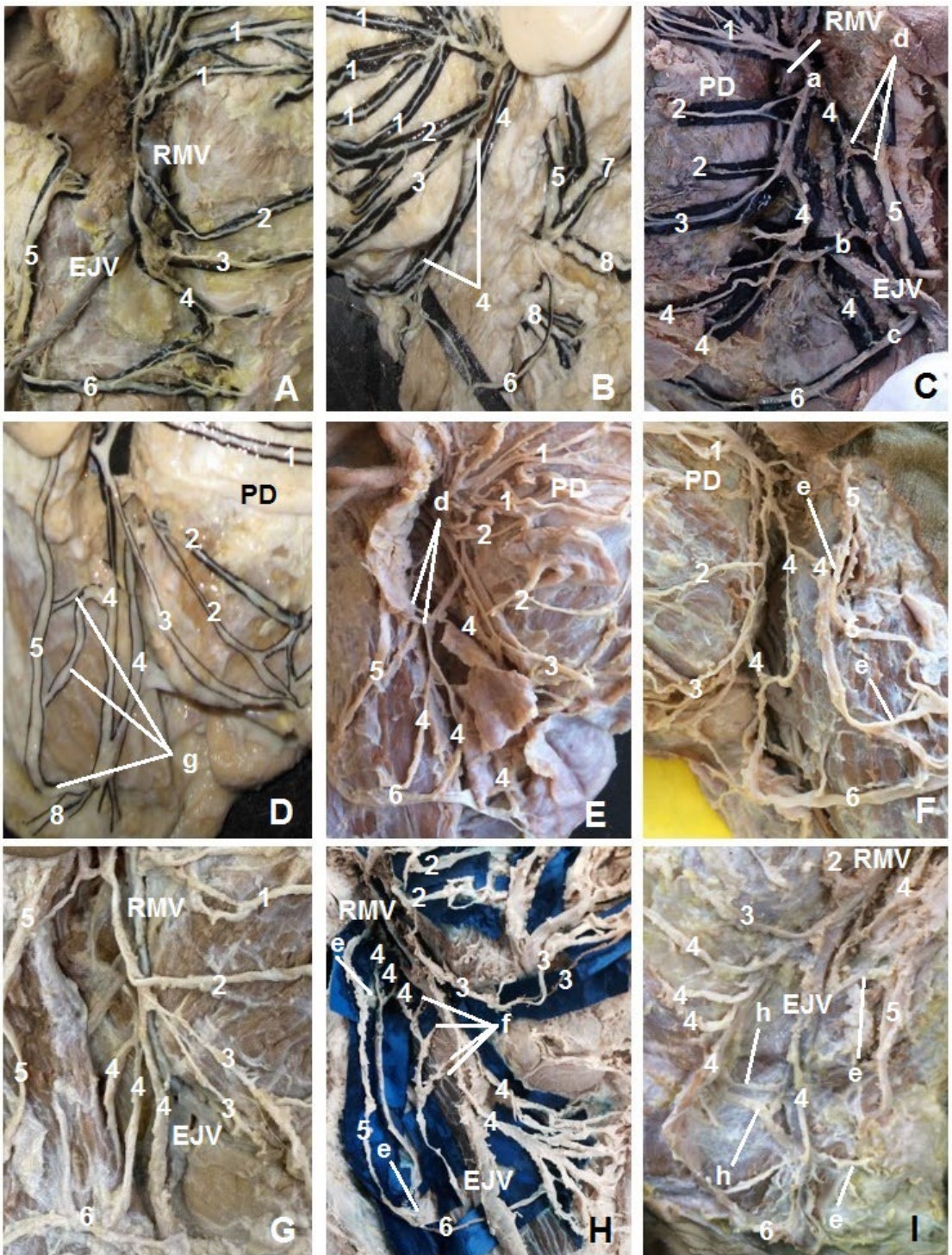

**Figure 4.** Connections and number variation in the cervical branch of the facial nerve. (**A**) Solitary cervical branch; (**B–D**) double cervical branch; (**E–G**) triple cervical branch; (**H,I**) multiple cervical branches. Here, 1—superior buccal branches; 2—inferior buccal branches; 3—marginal mandibular branch; 4—cervical branch; 5—greater auricular nerve (cervical plexus); 6—transverse cervical nerve (cervical plexus); 7—lesser occipital nerve (cervical plexus), 8—supraclavicular branches of the cervical plexus. (**C**)—a–c (superior, middle, and inferior connections between the cervical branches); (**C,E**)—d (double close connections between the cervical branch and greater auricular nerve); (**F,H,I**)—e (double distant connections between the cervical branch and greater auricular nerve); (**D**)—g (triple connections between the cervical branch and greater auricular nerve); (**I**)—h (double connections between the cervical branches); (**H**)—f (multiple connections between the cervical branches); RMV—retromandibular vein; EJV—external jugular vein; PD—parotid duct.

## 4. Discussion

Given the paramount importance of comprehending the innervation patterns of facial musculature in head and neck surgery, numerous studies have focused on characterizing facial nerve branches. Among these, the marginal mandibular nerve branch has received considerable attention for its role in innervating the lower lip musculature. However, despite efforts to safeguard this branch, reports indicate a substantial incidence of lower lip palsy, reaching up to 16% following neck dissection [8]. Current literature suggests that the cervical branch of the facial nerve is responsible for innervating the inferior labial depressor, elucidating the relatively elevated occurrence of partial lower lip palsy [8]. The functionality of this muscle holds significance in smile aesthetics, speech, and articulation.

Additionally, several studies have identified a notable frequency (3.6–40%) of communication branches between the marginal mandibular nerve and the cervical branch. This observation is of particular interest as these communication branches have been posited as potential avenues for motor recovery following facial nerve damage, offering protection against prolonged palsy [32]. Traditionally, the platysma incision is made 2 cm below the mandible, and the Hayes Martin maneuver, involving the ligation of the facial artery and vein, serves to shield the marginal mandibular nerve branch from harm [8]. Given the frequent documentation of lower lip asymmetry in the literature, likely attributed to cervical branch injury, anatomical studies investigating the variability of the cervical branch and factors influencing its trajectory are imperative.

According to Poutoglidis et al. [33], Type III is the most common branching pattern of the facial nerve, with a rate of 26.8%. This corresponds to the results obtained by this study, which showed the Type III branching pattern to prevail over the other types (20%). While the number and topographical variants of different facial branches have been described in detail, the cervical branch has yet to be studied as a peculiarity of the other facial nerve branches. According to Thomas et al. [34], the cervical branch is variable, and in iatrogenic injuries, along with facial aesthetic deformity, functional problems of oral competence and speech might occur. For example, the incidence of the transient dysfunction of the lower lip, in iatrogenic injury of the cervical branch, is about 3% [35].

Chowdhry et al. [10] determined that the cervical branch ramification point is $1.75 \pm 0.26$ cm below the intersection of the mento-mastoid line with the perpendicular line traced along the mandibular angle. In this study, measurements of the cervical branch landmarks were not taken, but according to the obtained results, the number, topography, and connections of the cervical branch are variable; thus, a precautious use of the projection landmarks is recommended.

The highest rate of a single cervical branch, 100%, was reported by Sinno et al. [28], and it was also reported in 94.71% of cases by Martínez Pascual et al. [29]. Ziarah et al. [23] found a single cervical branch in 80% of cases. In the present study, the number of cervical branches varied from 1 to 5 CB, and in 61.3% of cases, there was a single branch. Martínez Pascual et al. [29] showed two and three cervical branches in 2.6% of cases. Their results differ significantly from this study's, in which 2 CB were observed in 28% of cases and other variants in 10.7% of cases.

Ziarah et al. [23] mentioned that connections of the cervical branch with the transverse cervical and greater auricular nerves are frequent, but those with the marginal mandibular branch are rare. According to other data, the cervical branch is connected to the marginal mandibular branch in 12% of cases [24,25], and a twice-higher rate of 24.3% was obtained by Balagopal et al. [26]. In the current study, connections of the cervical branch with the mandibular branch were determined in 24% of cases, among which single, double, and multiple connections were observed. Connections of the cervical branch with the greater auricular nerve have also been reported by others [23,36]. This study showed the cervical branch's single, double, and triple connections with the greater auricular nerve in 38.7% of cases.

Connections of the cervical branch with the transverse cervical nerve of the cervical plexus are known and usually reported as common [37,38]. In the current study, in 100%

of cases, the cervical branch of the facial nerve was connected to the transverse cervical nerve. Variations of the cervical branch topography towards the retromandibular vein were reported by Zoulamoglou et al. [27], who described a superficial course of the cervical and mandibular branches of the facial nerve towards the retromandibular vein. In the present study, two cases (2.7%) of a bizarre topographical variation in the cervical branch were revealed when, in the proximity of the retromandibular vein, it divided into two branches that surrounded the retromandibular vein in a ring fashion and then continued as a single cervical branch behind the vein.

According to Sinno et al. [28], the cervical branch extends its course toward the medial margin of the platysma muscle, located below the thyroid cartilage. In this study, that course was characteristic only for the hemifaces with a single cervical branch. However, their terminal divisions were distinguished above and below the thyroid cartilage in cases with more than one cervical branch.

The highest level of the atypical branching variants was observed in Type II according to the Davis classification and the lowest for Types V and VI. The cervical branch of the facial nerve was subjected to a high degree of variability in all the examined categories. Its number varied from 1 to 5 CB, among which 89.3% were cases with one or two cervical branches. The mean number of cervical branches was higher in male individuals and on the right hemifaces. The lowest and similar mean number of 1.4 CB was characteristic for Types II, III, and IV, and the highest number of 2.0 CB was revealed in Type V. The lowest number variation was determined in the mesocephalic type, with a mean of 1.5 CB, and the highest in the brachycephalic type, with a mean of 1.8 CB. The cervical branch of the facial nerve was connected to the transverse cervical nerve in 100% of cases, to the greater auricular in 38.7% of cases, and to the marginal mandibular branch in 24% of cases. Five types of connections between the cervical branch of the facial nerve and the greater auricular nerve were observed. There were determined to be single, double, and triple connections among cervical branches.

Shortcomings of the study might include the ethnic homogeneity of the specimens, which could explain the different outcomes of human anatomical studies in general. Furthermore, the number of specimens was not equally distributed between the genders; here, gender effects need to be interpreted cautiously.

According to the published data, the extended myotomy of the platysma muscle is safe at a minimum of 3 cm below the mandibular margin [39] or 5 cm below the mandibular angle [40]. In the performed dissections, the distance between the cervical branch's course and the sternocleidomastoid muscle's anterior margin was at least 1.0 to 1.5 cm, even in cases with double, triple, and multiple cervical branches. Despite the reported partial innervation of the lower lip by the cervical branch of the facial nerve, the dominant branch innervating the inferior labial depressor lies $2 \pm 0.5$ cm below the angle of the mandible and travels perpendicular to the mento-mastoid process line [8]. Thus, the mentioned area can be considered a relatively safe zone for platysma muscle dissection in neck dissection, salivary gland surgery, rhytidectomy, and other surgical procedures.

## 5. Conclusions

Without a sufficient method to display facial nerve branching patterns, knowledge of the most likely course of those branches is of great importance. Based on this study, typical branching patterns may be assigned to either patient according to the gender, side, cephalic index, and branching classification of the facial nerve. Nevertheless, cautious dissection must be performed during head and neck surgery due to the great inter- and intraindividual variability of the cervical branches' route and the number of branches.

**Author Contributions:** Conceptualization, A.B., V.P., S.L. and P.W.K.; methodology, A.B., I.C., Z.Z., S.V., S.L. and P.W.K.; validation, A.B., V.P., S.L., P.W.K. and D.H.; formal analysis, A.B., V.P., I.C., Z.Z., S.V. and S.L.; investigation, A.B.; V.P., I.C., Z.Z., S.V. and S.L.; resources, A.B., V.P. and S.L.; data curation, A.B., V.P., I.C., S.V., Z.Z. and S.L.; writing—original draft preparation, A.B., V.P., D.H. and P.W.K.; writing—review and editing, A.B., V.P., I.C., Z.Z., S.V., S.L., D.H. and P.W.K.; visualization, A.B., V.P., S.L. and D.H.; supervision, A.B., V.P., P.W.K. and S.L.; project administration, A.B., V.P., P.W.K. and S.L. All authors have read and agreed to the published version of the manuscript.

**Funding:** This research received no specific grant from any funding agency in the public, commercial, or not-for-profit sectors.

**Institutional Review Board Statement:** The research project was approved by the Ethics Committee of Nicolae Testemitanu State University of Medicine and Pharmacy in the Republic of Moldova (minute No. 1 of 19 September 2014). The study was conducted in full accordance with the Declaration of Helsinki.

**Informed Consent Statement:** Informed consent was obtained from all subjects involved in the study.

**Data Availability Statement:** The datasets generated during and/or analyzed used in this manuscript are available from the corresponding author upon reasonable request.

**Conflicts of Interest:** The authors declare that the research was conducted in the absence of any commercial or financial relationships that could be construed as potential conflicts of interest.

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
