# Peer review of "Variability of the Cervical Branch Depending on the Facial Nerve Branching Pattern and Anthropometric Type of the Head"

_2035-8377, doi:10.3390/neurolint16010007_

Round 1

Reviewer 1 Report

Comments and Suggestions for Authors

A well done anatomical study. Would you please expand on why you analysed the brachycephalic, dolicocephalic anthropometry in the methodology. How is this relevant to the facial nerve branching patterns. Also your dissections are extensive and can be quite confusing to look after. I would suggest streamlining these with associated line diagrams. 

Comments on the Quality of English Language

Average. It can be edited to make for a smoother read. 

Reviewer 2 Report

Comments and Suggestions for Authors

Thank you for the opportunity to review your interesting article.

This study is the theme of variability of the cervical branch of the facial nerve.

It is important and useful to study precious anatomy of facial nerve when we do surgery of parotid tumor and neck dissection.

As authors presented L47-48, marginal mandibular nerve damage in the case of neck dissection is functional problem for the patients. But the cervical branches are resected without functional problems. Please include the introduction of meaning for preserving cervical branches. 

Figure 1: The type are difficult to find in the figures (Type I,II,III…). Please include the clear type letters in the figure. What were the meaning of Type I-VI 13%-6%? Please include the explanation.

Figure 2: Are the greater auricular nerve and lesser occipital nerve sensory nerve?

Figure 3: The order of picture A,B,C…. and figure legends are different, and could not understand easily. Please modify the figure presentation.

Material and Methods

Davis’s classification was not clear for readers. Please include the explanation of classification.

Discussion

L198-201: Is this article about the marginal mandibular branch? Usually cervical branch resection dose not occurs functional problems.

Comments on the Quality of English Language

Minor editing of English language required.

Reviewer 3 Report

Comments and Suggestions for Authors

This is a useful study which will be of interest to surgeons working in the neck.

I was interested to see that Figure 1 shows only slight and subtle differences in the cervical branch patters , even if the nerve as a whole showed wide variation. The CB looks the same for types II and III.

I don't think the authors make this point.

Table 1 has small numbers. Please check if your Chi square test needs Yates correction for small samples.

Line 240 - David or Davis ?

I would like to see some discussion of the surgical importance , or otherwise, of the CB for patients. Ideally this should be at the start of the Discussion. I was taught that the CB is unimportant and OK to sacrifice in neck surgery . Has opinion changed ?

Reference 5 has the title in block brackets - why ?

Apart from these comments, I am happy with this paper.

Round 2

Reviewer 2 Report

Comments and Suggestions for Authors

This article is acceptable for publication.

Comments on the Quality of English Language

Minor editing required.